# Multimodal Sentiment Analysis Based on Causal Reasoning

## Abstract

With the rapid development of multimedia, the shift from unimodal textual sentiment analysis to multimodal image-text sentiment analysis has obtained academic and industrial attention in recent years. However, multimodal sentiment analysis is affected by unimodal data bias, e.g., text sentiment is misleading due to explicit sentiment semantic, leading to low accuracy in the final sentiment classification. In this paper, we propose a novel **C**ounter**F**actual **M**ultimodal **S**entiment **A**nalysis framework (CF-MSA) using causal counterfactual inference to construct multimodal sentiment causal inference. CF-MSA mitigates the direct effect from unimodal bias and ensures heterogeneity across modalities by differentiating the treatment variables between modalities. In addition, considering the information complementarity and bias differences between modalities, we propose a new optimisation objective to effectively integrate different modalities and reduce the inherent bias from each modality. Experimental results on two public datasets, MVSA-Single and MVSA-Multiple, demonstrate that the proposed CF-MSA has superior debiasing capability and achieves new state-of-the-art performances. We will release the code and datasets to facilitate future research.

## 1 Introduction

Sentiment analysis has been the fundamental research in the field of artificial intelligence. Early research on sentiment analysis (Wang et al., 2016; Bhange & Kasliwal, 2020) mainly focus on a single text modality, making it difficult to deal with the complex multimodal scenarios and thus to accurately capture the accurate users' sentiments. Benefited from the development of information technology and the widespread popularity of mobile device, a large amount of multimodal data such as text and image can be uploaded and disseminated without delay. Multimoal data is also verified to enlarge broader real-world application prospects, such as personalized recommendations (Gao et al., 2016), sentiment cross-modal retrieval (Xu & Li, 2023) and user depression estimation (Sun et al., 2022), etc. Such habits and prospects prompt the academic and industrial institutions to shift from unimodal sentiment analysis to the multimodal one (Gandhi et al., 2023). attracts increasing research and industrial attention, as Therefore, it has become a research trend to shift from unimodal sentiment analysis to multimodal sentiment analysis.

In recent years, with the booming development of deep learning techniques, many works have extensively penetrated the field of multimodal sentiment analysis (Hu et al., 2022; Shi & Huang, 2023; Jiang et al., 2023; Xu & Mao, 2017; Zhu et al., 2023a; Li et al., 2024). The mainstream of them, including multimodal large language model (MLLM) Achiam et al. (2023); Dubey et al. (2024) mainly relies on the joint learning of multimodal data, i.e., predicting a unified sentiment by fusing the multimodal features. Despite the promising performances, such mainstream mixes up the hidden multimodal features without tracking the causal links to ensure the actual semantic elements to resolve the sentiment judgment. For example, Zhu et al. (2023a) proposes an image-text multimodal interaction network for sentiment analysis, exploring the inter-modal alignment during modality fusion. However, they still only focus on inter-modal interactions and ignore the effects of causality between multimodal features which leads to misleadingness of the sentiment expressions in the wrong modality, i.e. modality bias. In fact, modality bias is a common phenomenon in multimodal sentiment prediction models due to different semantic expressivity of different modalities, such as explicit textual sentiment keywords and shallow visual semantic. For example, Figure1 shows that the red-font text causes the higher negative possibility while cat causes the higher positive possibil-

ity. Therefore, *how to perform unbiased inference during the training process with modality bias remains a major challenge for multimodal sentiment prediction.*

In real world, humans are able to rely on causal reasoning to effectively distinguish good bias from bad bias and thus make more unbiased predictions. Inspired by this, we introduce counterfactual causal reasoning to help the model eliminate individual modal biases, thereby identifying the main modality causal effects that truly determine sentiment classification. Specifically, counterfactual causal inference allows the model to think: "Would the same prediction have been made if the particular modality information had not been seen?" With this inference pattern, the model can simulate scenarios that weed out the effects of individual modalities, helping the model to identify the main causal effects in multimodal scenarios that actually determine affective categorization, and thus make more unbiased predictions. To do this, we formulated the modality bias in the multimodal sentiment prediction as the direct causal effect of each modality on labeling, and mitigated the modality bias by subtracting the direct modality effect from the total causal effect.

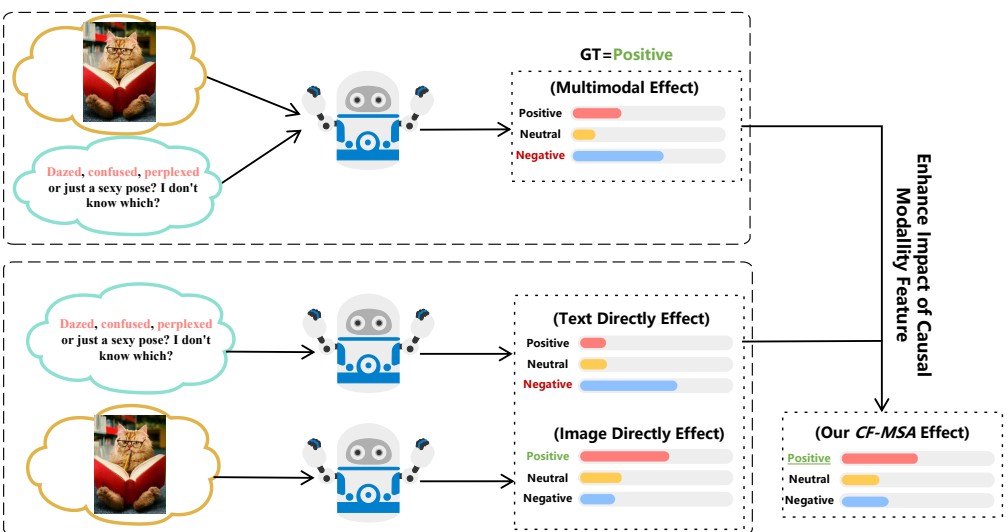

Figure 1: The upper part illustrates traditional likelihood-based multimodal sentiment prediction, where models rely on biased dominant modalities (e.g., explicit emotional words in text). It contains certainly misleading. The lower part demonstrates our counterfactual reasoning approach, which mitigates such bias by analyzing the impact of missing modalities, leading to more unbiased multimodal predictions.

As shown in Figure1, the upper part demonstrates the traditional likelihood-based biased prediction process, where these models rely on the multimodal joint prediction under biased training. It leads to over-reliance on a particular modality with high sentimental properties in prediction, such as the explicit sentiment words in text, and ignores the potential information of other modalities. This type of learning is prone to modality bias, especially when containing the explicit semantic information, and the model is more inclined to bias the prediction towards the dominant modality. In contrast, the following counterfactual reasoning analyzes the prediction results in the absence of the influence of a single modality by simulating the case where the modality is missing, and by comparing the difference between the two cases we can determine the influence of a single modality on the prediction, thus eliminating the bias caused by the single modality. We use the generalized conventional model and unimodal branching to train the ensemble model in the model training phase, and in the testing phase, the reasoning is done through counterfactual thinking, aiming to make the model focus more on the unbiased multimodal fusion information and thus make unbiased predictions. Experimental results show that the CF-MSA method outperforms current state-of-the-art methods on the MVSA-Single and -Multiple datasets.

We summarize our contributions as follows: (I) We are the first to introduce the causal effects into multimodal sentiment analysis framework. By innovatively proposing a counterfactual multimodal sentiment analysis framework, the bias introduced by modality is taken into account and a more comprehensive prediction is realized. (II) The causal relations used in our proposed CF-MSA framework

are generic and can be applied to different multimodal sentiment analysis task architectures and fusion strategies. (III) The method provides a new perspective on the multimodal sentiment analysis debiasing task and achieves superior results on MVSA-Single and MVSA-Multiple datasets.

## 2 RELATED WORKS

In recent years, multimodal sentiment analysis has become an important research direction in the field of sentiment computing (Gandhi et al., 2023). Studies (Zhu et al., 2023a) show that fusing features from different modalities can significantly improve the performance of sentiment classification, especially in the analysis of unstructured data such as social media. Therefore, As technology has progressed, sentiment analysis has expanded from single-text analysis to multimodal sentiment analysis, which more comprehensively captures subtle changes in sentiment and provides more accurate sentiment predictions. Early work in this area has focused on developing methods to better fuse data from different modalities. For example, Xu & Mao (2017) proposed MultiSentiNet , a deep semantic network for multimodal sentiment analysis in 2017. The model improves the accuracy of sentiment classification by extracting object and scene semantic features from images and combining them with sentiment information from text. De Toledo and Marcacini introduce a joint fine-tuning-based transfer learning approach (de Toledo & Marcacini, 2022) for multimodal sentiment analysis. Their approach combines pre-trained unimodal models (e.g., text and images) and achieves a multimodal representation by jointly optimizing these models in a fine-tuning step, which requires significantly less computational resources compared to complex models such as CLIP (Radford & et al., 2021). Rajan et al. (2022) explore the relationship between Cross-Attention and Self-Attention in their study. Meanwhile, Zhu et al. (2023b) introduce a multimodal approach for multi-level emotion classification. Their work further improves the accuracy and robustness of emotion classification by combining multiple pre-trained unimodal models and effectively integrating them in the multimodal space.

However, these methods do not go further in the study of the causal relationships implied in the task. In data-driven model learning, due to modality bias, the model cannot distinguish the impact of unimodal information from that of multimodal fused information, which leads to the model learning spurious correlations. Aware of this drawback in many existing models, counterfactual thinking and causal inference have received attention in related fields in recent years, and have provided solutions to eliminate bias (Tang et al., 2020), enhancing model robustness (Louizos et al., 2017), and for supporting complex decision-making applications (Gencoglu & Gruber, 2020; Liu et al., 2023a;b) have provided solutions to these problems. In the multimodal context, researchers and scholars have applied causal inference to various domains, including video question answering (Zang et al., 2023), image caption generation, visual tasks (Wang et al., 2020; 2021), and the design of various visual-language reasoning tasks (Yang et al., 2021; Rao et al., 2021). Inspired by this idea, this paper proposes a counterfactual multimodal sentiment analysis (CF-MSA) model that introduces causal reasoning to mitigate the negative impact of modality bias on performance, thereby achieving more accurate multimodal sentiment analysis.

## 3 METHODS

In this section, we introduce causal relationships and counterfactual reasoning in multimodal sentiment analysis tasks in Section 3.1. More detailed information about counterfactual causal reasoning can be found in Appendix A. Then, in Section 3.2, we introduce how to introduce causal reasoning into experiments on multimodal sentiment analysis.

### 3.1 CAUSE-EFFECT

As shown in Figure 2, text and image can not only influence sentiment labels through direct unimodal paths (e.g. Text $\rightarrow Y$ or Image $\rightarrow Y$), but can also act indirectly on sentiment labels through multimodal paths. Such multimodal paths typically involve the intermediate action of multimodal information mediator $K$, i.e., Text and Image together influence $K$, and then $K$ influences $Y$ (Text, Image $\rightarrow K \rightarrow Y$).

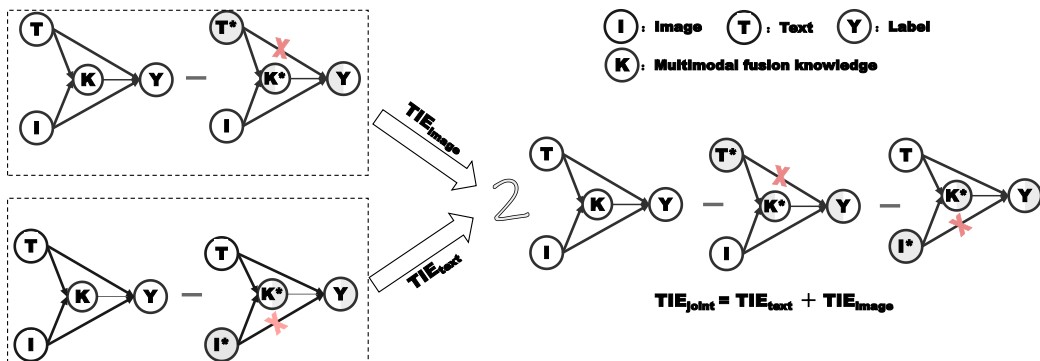

Figure 2: The analysis of the unimodal $TIE$ effect for image and text is shown on the left, where $T^*$ and $I^*$ indicate that text and image information is masked respectively, and the red crosses indicate that the path is blocked. In this case, the model relies only on unimodal information for sentiment prediction. The joint text-image effect ($TIE_{\text{joint}}$) is shown on the right.

In counterfactual notation, in the original traditional causality, for a given pair of text-image pairs, the resultant labeling score can be found as follows:

$$Y_{t,i} = Y(y; T = t, I = i) \tag{1}$$

For simplicity, $y$ is omitted in this study, i.e., $Y_{t,i} = Y(T = t, I = i)$. Similarly, the counterfactual notation for $K$ is denoted as $K_{t,i} = K(T = t, I = i)$.

Therefore, there are three paths in the counterfactual causal sentiment analysis that directly influence the decision of final sentiment polarity $Y$, namely $T \rightarrow Y$, $I \rightarrow Y$, and $K \rightarrow Y$, respectively. To this end, we rewrite $Y_{t,i}$ as a function $Z_{t,i,k}$ of $T$, $I$, and $K$:

$$Y_{t,i} = Z_{t,i,k} = Z(T = t, I = i, K = k) \tag{2}$$

If we want to mask the image influence in counterfactual sentiment analysis to estimate the causal effect of text on the outcome, $K$ will reach a value of $k^*$ when $T$ is set to $t$ and $I$ to $i^*$. Since the response of the mediator $K$ to the input is blocked, the model can only rely on the given text for decision making. In this case, $Y_{t,i^*}$ is represented as follows:

$$Y_{t,i^*} = Z_{t,i^*,k^*} = Z(T = t, I = i^*, K = k^*) \tag{3}$$

Similarly, if in counterfactual sentiment analysis one wants to mask the text influence to estimate the causal influence of the image on the outcome, in this case, $Y_{t^*,i}$ is represented as follows:

$$Y_{t^*,i} = Z_{t^*,i,k^*} = Z(T = t^*, I = i, K = k^*) \tag{4}$$

In order to quantify the effects brought about by the text, the present study defines the total effect (TE) based on the sentiment $Y = y$ for a specific text $T = t$ and image $I = i$ condition. The total effect can be expressed as:

$$TE = Y_{t,i} - Y_{t^*,i^*} = Z_{t,i,k} - Z_{t^*,i^*,k^*} \tag{5}$$

The natural direct effect (NDE) represents the direct effect of text on sentiment with a fixed intermediate variable $K$. The formula is calculated as follows:

$$NDE = Z_{t,i^*,k^*} - Z_{t^*,i^*,k^*} \tag{6}$$

The value of NDE can to some extent be used to assess the direct effect that text brings to the labeling of sentiment polarity. The final study by subtracting NDE from TE , can be used to assess the modeling effect of reducing text bias, the calculation of total indirect effect(TIE) in multimodal sentiment analysis is represented as follows:

$$TIE = TE - NDE = Z_{t,i,k} - Z_{t,i^*,k^*} \tag{7}$$

This study selects the answer that maximizes the TIE for sentiment inference, which is significantly different from the traditional strategy based on the posterior probability $P(y \mid t, i)$.

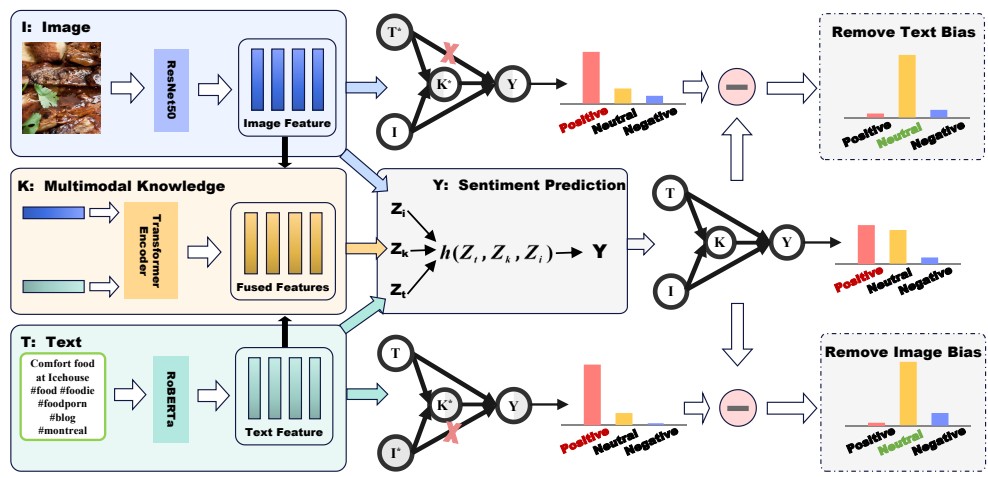

(a) Biased Training of MSA Framework     (b) Unbiased Causal CounterFactual Inference

Figure 3: CF-MSA model training consists of three main branches: the text branch ($Z_t$), the image branch ($Z_i$), and the text-image integration branch ($z_k$). The testing phase uses causal counterfactual inference to make unbiased sentiment label predictions.

The total indirect effect (TIE) is calculated as subtracting the direct effects of the separate text and separate image from the total effect. This can be accomplished in the following two steps:

(1) Calculate the indirect effect of the text:

$$TIE_{\text{text}} = Z_{t,i,k} - Z_{t,i^*,k^*} \tag{8}$$

where $Z_{t,i,k}$ is the expected value of emotion when considering the joint effect of text and image, while $Z_{t,i^*,k^*}$ is the expected value of emotion when fixing the image and changing the text.

(2) Calculate the indirect effect of the image:

$$TIE_{\text{image}} = Z_{t,i,k} - Z_{i,t^*,k^*} \tag{9}$$

where $Z_{i,t^*,k^*}$ is the expected value of emotion when fixing the text and changing the image. The joint total indirect effect (TIE) is a composite effect that adds the above two indirect effects so as to capture the combined effect of text and image jointly influencing sentiment polarity through mediating variables, and it is shown in Figure 2:

$$TIE_{\text{joint}} = TIE_{\text{text}} + TIE_{\text{image}} \tag{10}$$

## 3.2 IMPLEMENTATION

The study further proposes a counterfactual multimodal sentiment analysis (CF-MSA) framework based on causal reasoning. As shown in Figure 3, sentiment analysis is accomplished by fusing three independent model branches: pure text, pure image, and text-image integration, which process different input modalities respectively, and combine them through a fusion function $h$ to form the final sentiment polarity prediction.

**Parameterization and model definition.** The model used in the experiments is parameterized as follows, based on three branching models $F_T$, $F_I$, $F_{IT}$ and a fusion function $h$:

$$Z_t = F_T(t), \quad Z_i = F_I(i), \quad Z_k = F_{IT}(i,t), \quad Z_{t,i,k} = h(Z_t, Z_i, Z_k) \tag{11}$$

where $F_T$ is a pure text branch dealing with the mapping from text to emotion (i.e. $T \rightarrow Y$), $F_I$ is a pure image branch dealing with the mapping from image to emotion (i.e. $I \rightarrow Y$), and $F_{IT}$ is a synthesized branch of text-image synthesis (i.e. $I, T \rightarrow K \rightarrow Y$). The output scores are fused by the function $h$ to obtain the final score $Z_{t,i,k}$.

**Intermodal discrepancy treatment.** Under processing conditions, the image or text signal is masked, i.e., no image ($i$) or text ($t$) is provided. In order to effectively process input uncertainty, this study designs an intermodal discrepancy processing that can adjust its behavior in the presence of different modal input defects. The experimental method is to set specific parameters ($c1, c2, c3, c4$) for each different input defect situation to simulate the prediction behavior in different situations. Therefore, in the absence of input, the score is expressed as:

$$Z_t = \begin{cases} z_t = F_T(t) & \text{if } T = t \\ z_t^* = c1 & \text{if } T = \emptyset \end{cases} \tag{12}$$

$$Z_i = \begin{cases} z_i = F_I(i) & \text{if } I = i \\ z_i^* = c2 & \text{if } I = \emptyset \end{cases} \tag{13}$$

$$Z_k = \begin{cases} z_k = F_{IT}(i, t) & \text{if } I = i \text{ and } T = t \\ z_k^* = c3 & \text{if } I = i \text{ or } T = \emptyset \\ z_k^* = c4 & \text{if } I = \emptyset \text{ or } T = t \end{cases} \tag{14}$$

where $c = (c1, c2, c3, c4)$ denotes the learnable parameters, and it is assumed that they satisfy a uniform distribution. There are two reasons for using the uniform distribution assumption in this study; for human beings, if they are completely unaware of the exact treatment, including text or image, people would want to make a bold guess, and here the parameters denote a random probability that should be consistent with the uniform distribution assumption. The second reason is because the subsequent $c$ parameters are in turn designed to estimate the NDE, and the assumption of uniform distribution ensures the safety of the estimation process.

**Integration strategy.** In order to fuse the output scores $Z_t$, $Z_i$, and $Z_k$ from different models to obtain $Z_{t,i,k}$, a SUM nonlinear fusion strategy is used:

$$(\text{SUM}) \quad h(Z_t, Z_i, Z_k) = \log \sigma(Z_{\text{SUM}}), \quad \text{where} \quad Z_{\text{SUM}} = Z_t + Z_i + Z_k \tag{15}$$

**Intermodal bias mutual elimination optimisation objective.** The possible extremes of bias in image and text are considered, and the inherent tendency of single modal data often negatively affects the efficacy and accuracy of the overall model. For this reason, this study also proposes an innovative intermodal bias mutual elimination optimisation objective, which centers on eliminating bias by optimizing the intermodal probability distribution to maximize the use of information and complement each other, so as to improve the performance and accuracy of the overall model. Simply understood, it is to pull the two extremes toward a more balanced probability distribution.

The specific approach in the experiment is to introduce a new loss function $\mathcal{L}_{ti}$, which minimizes the difference between the probability distributions of text and image patterns by minimizing the Kullback-Leibler divergence between them. This loss function is defined as follows:

$$\mathcal{L}_{ti} = \frac{1}{|Y|} \sum_{y \in Y} p(y \mid t, i^*, k^*) \log \frac{p(y \mid t, i^*, k^*)}{p(y \mid t^*, i, k^*)} + \frac{1}{|Y|} \sum_{y \in Y} p(y \mid t^*, i, k^*) \log \frac{p(y \mid t^*, i, k^*)}{p(y \mid t, i^*, k^*)} \tag{16}$$

**The learnable parameter c.** The learnable parameter $c$, which controls the sharpness of the $Z_{t,i^*,k^*}$ and $Z_{t^*,i,k^*}$ distributions, assumes that the sharpness of the NDE should be similar to that of the TE. Otherwise, the study concluded that inappropriate $c$ could lead to TIE being dominated by either TE or NDE. Therefore, the experiment uses the Kullback-Leibler divergence to estimate $c$, which is expressed by the simplified optimization objective function as follows::

$$\mathcal{L}_{kl1} = \frac{1}{|Y|} \sum_{y \in Y} -p(y \mid t, i, k) \log p(y \mid t, i^*, k^*) \tag{17}$$

$$\mathcal{L}_{kl2} = \frac{1}{|Y|} \sum_{y \in Y} -p(y \mid t, i, k) \log p(y \mid t^*, i, k^*) \tag{18}$$

$$\mathcal{L}_{kl} = \mathcal{L}_{kl1} + \mathcal{L}_{kl2} \tag{19}$$

where $p(y \mid t, i, k)$ denotes the probability of the output of the softmax function based on $Z_{t,i,k}$, while $p(y \mid t, i^*, k^*)$ and $p(y \mid t^*, i, k^*)$ denote the probability of the model output when the processing conditions are modified respectively. Only $c$ is updated when minimizing $\mathcal{L}_{kl}$ and $\mathcal{L}_{t_i}$.

Table 1: Statistical summary of the processed MVSA-Single and MVSA-Multiple datasets.

| Dataset | Positive | Neutral | Negative | Total |
|---|---|---|---|---|
| MSVA-Single | 2683 | 470 | 1358 | 4511 |
| MSVA-Multiple | 9327 | 1091 | 6359 | 16779 |

**Training.** In the training phase, the main goal of model optimization is to minimize the cross-entropy loss for a given triad $(t, i, y)$, where $y$ is the sentiment polarity label of the text-image pair $(t, i)$, and then to optimize the following loss function to tune the model branches:

$$\mathcal{L}_{cls} = \mathcal{L}_{\mathcal{ITY}}(i, t, y) + \mathcal{L}_{\mathcal{TY}}(t, y) + \mathcal{L}_{\mathcal{IY}}(i, y) \tag{20}$$

where $\mathcal{L}_{\mathcal{ITY}}$, $\mathcal{L}_{\mathcal{TY}}$, and $\mathcal{L}_{\mathcal{IY}}$ are the values of $Z_{t,i,k}$, $Z_t$, and $Z_i$ respectively.

The final loss function is:

$$\mathcal{L} = \sum_{(t,i,y) \in D} \mathcal{L}_{cls} + \mathcal{L}_{kl} + \mathcal{L}_{ti} \tag{21}$$

**Inference.** The experiments are conducted using debiased causal effects for inference, which take into account the text bias implemented as follows:

$$TIE = TE - NDE = Z_{t,i,k} - Z_{t,i^*,k^*} \quad = h(z_t, z_i, z_k) - h(z_t, z_i^*, z_k^*) \tag{22}$$

Similarly, consider the image bias implemented as follows:

$$TIE = TE - NDE = Z_{t,i,k} - Z_{t^*,i,k^*} \quad = h(z_t, z_i, z_k) - h(z_t^*, z_i, z_k^*) \tag{23}$$

Combining the two yields a text-image bias implementation

$$TIE = TE - NDE = 2 \times Z_{t,i,k} - Z_{t,i^*,k^*} - Z_{t^*,i,k^*}$$
$$= 2 \times h(z_t, z_i, z_k) - h(z_t, z_i^*, z_k^*) - h(z_t^*, z_i, z_k^*) \tag{24}$$

where the coefficient 2 reflects the inclusion of both text and image biases in the computation, ensuring that the effects of both modalities are equally represented in the final result.

## 4 EXPERIMENTS

### 4.1 DATASETS AND SETTINGS

**Datasets and Evaluation Protocols.** We provide evaluations on the widely used datasets, i.e. he MVSA-Single dataset and the MVSA-Multiple dataset. The MVSA-Single dataset contains 5,129 image-text pairs collected from Twitter, and each image corresponds to only one text annotation. The MVSA-Multiple dataset contains 19,600 image-text pairs, where each pair is labelled by three annotators for rating. In particular, we preprocessed the two datasets according to Xu & Mao (2017) by removing image-text pairs with inconsistent modality labels. In this way, we obtained more accurate multimodal sentiment data. The detailed statistics of the two datasets are shown in Table 1 Quantitative performances of all methods are evaluated by the Accuracy and F1 scores.

**Implementation Settings.** In this section, we briefly describe the key settings of our model implementation and training process. The model consists of a BERT-based text encoder, a ResNet-based image encoder, and a multi-head attention fusion module. The AdamW optimizer was used for training with separate learning rates for different components. The model was trained for 20 epochs with a combination of cross-entropy and KL divergence loss. For detailed implementation settings, including the specific hyperparameters, optimizer configurations, and training procedures, please refer to the Appendix B.

### 4.2 BASELINE AND STATE-OF-THE-ART METHODS

We compare the proposed CF-MSA with a biased multimodal sentiment prediction model, containing the text-image bias, i.e. $I \rightarrow Y$ and $T \rightarrow Y$ shown in Figure 3. Based on the baseline, we can explore the effects of multiple modality biases for the final sentiment predictions and

Table 2: Comparisons of experimental results on the MVSA-Single and MVSA-Multiple datasets.

| Method | Condition | MVSA-Single | | MVSA-Multiple | |
|---|---|---|---|---|---|
| | | Accuracy | F1 | Accuracy | F1 |
| OTE | Multimodal Biased | 73.24 | 72.93 | 65.40 | 64.78 |
| CMAC | Multimodal Biased | 68.55 | 64.27 | 64.44 | 61.58 |
| HSTEC | Multimodal Biased | 71.60 | 67.49 | 63.25 | 60.27 |
| NaiveCat | Multimodal Biased | 72.46 | 71.79 | 65.52 | 65.65 |
| NaiveCombine | Multimodal Biased | 66.02 | 65.93 | 55.81 | 49.27 |
| Baseline | Multimodal Biased | 73.63 | 70.90 | 65.63 | 64.78 |
| **CF-MSA (Ours)** | Removing Text Bias | **76.17** | **74.54** | **67.12** | 64.92 |
| | Removing Image Bias | 73.24 | 72.32 | 66.23 | 64.53 |
| | Removing Text-Image Bias | 74.80 | 74.13 | 65.75 | **64.99** |

Table 3: Performance comparisons when considering different modality biases on MVSA datasets.

| Method | MVSA-Single | | MVSA-Multiple | |
|---|---|---|---|---|
| | Accuracy | F1 | Accuracy | F1 |
| **Removing Text Bias (w/o $T \rightarrow Y$):** | | | | |
| CF-CMAC | $74.80_{(\uparrow1.56)}$ | $73.48_{(\uparrow0.55)}$ | $67.12_{(\uparrow1.60)}$ | $65.55_{(\downarrow0.10)}$ |
| CF-HSTEC | $76.17_{(\uparrow2.93)}$ | $74.97_{(\uparrow2.04)}$ | $65.40_{(\downarrow0.12)}$ | $64.49_{(\downarrow1.16)}$ |
| CF-NaiveCat | $75.78_{(\uparrow2.54)}$ | $74.46_{(\uparrow1.53)}$ | $66.52_{(\uparrow1.00)}$ | $65.26_{(\downarrow0.39)}$ |
| **Removing Image Bias (w/o $I \rightarrow Y$):** | | | | |
| CF-CMAC | $73.24_{(\uparrow0.00)}$ | $72.77_{(\downarrow0.16)}$ | $63.13_{(\downarrow2.39)}$ | $61.45_{(\downarrow4.20)}$ |
| CF-HSTEC | $74.61_{(\uparrow1.37)}$ | $73.64_{(\uparrow0.71)}$ | $66.11_{(\uparrow0.59)}$ | $65.09_{(\downarrow0.56)}$ |
| CF-NaiveCat | $73.24_{(\uparrow0.00)}$ | $72.64_{(\downarrow0.29)}$ | $66.41_{(\uparrow0.89)}$ | $66.08_{(\uparrow0.43)}$ |
| **Removing Text-Image Bias (w/o $I \rightarrow Y$ and $T \rightarrow Y$):** | | | | |
| CF-CMAC | $73.24_{(\uparrow0.00)}$ | $70.87_{(\downarrow2.06)}$ | $65.16_{(\downarrow0.36)}$ | $65.16_{(\downarrow0.49)}$ |
| CF-HSTEC | $73.83_{(\uparrow0.59)}$ | $71.45_{(\downarrow1.48)}$ | $66.83_{(\uparrow1.31)}$ | $65.80_{(\uparrow0.15)}$ |
| CF-NaiveCat | $73.44_{(\uparrow0.20)}$ | $71.49_{(\downarrow1.44)}$ | $65.46_{(\downarrow0.06)}$ | $64.12_{(\downarrow1.53)}$ |

prove the effectiveness of the novel counterfactual causal reasoning for multimodal sentiment analysis. In addition, we also compare the state-of-the-art methods, i.e., OTE (de Toledo & Marcacini, 2022), CMCA (Rajan et al., 2022),HSTEC (Rajan et al., 2022), NaiveCat (de Toledo & Marcacini, 2022), and NaiveCombine (de Toledo & Marcacini, 2022) to demonstrate the competitiveness of our method for sentiment prediction. Specifically, NaiveCat concatenates the text and image features extracted by BERT and ResNet, respectively, and then passes them through a classifier. In contrast, NaiveCombine is a combination of ResNet and BERT, where the model generates separate probability vectors for text and image features, and then combines them by summing the probability vectors before applying the softmax function to obtain the final prediction. The implementations of all these baselines can be found in the repository provided by Zheng (2022).

## 4.3 QUANTITATIVE RESULTS

We perform extensive experiments in Table 2 to compare our CF-MSA with mainstream multimodal sentiment analysis methods with multimodal biases on two MVSA-Single and -Multiple datasets in terms of accuracy (ACC) and F1 scores.

When comparing the models proposed with the baseline model, the CF-MSA model shows better performance for sentiment analysis with different unbiased-modality training, especially when dealing with text bias. Spefically, when removing the text bias to train the prediction model, our CF-MSA achieves 2.93% and 1.72% improvements in terms of ACC on two datasets. Furthermore, our unbiased CF-MSA models achieve new state-of-the-art performance in sentiment prediction. For ex-

Table 4: Comparisons of different methods with and without our proposed $\mathbf{L}_{t_i}$ on MVSA datasets

| Methods | MVSA-Single | | MVSA-Multiple | |
|---|---|---|---|---|
| | **Accuracy** | **F1** | **Accuracy** | **F1** |
| OTE | 72.66 | 72.00 | 64.03 | 63.96 |
| + $\mathbf{L}_{t_i}$ | 74.80$_{(\uparrow 2.14)}$ | 74.13$_{(\uparrow 2.13)}$ | 65.75$_{(\uparrow 1.72)}$ | 64.99$_{(\uparrow 1.03)}$ |
| CMAC | 73.24 | 71.31 | 65.81 | 64.74 |
| + $\mathbf{L}_{t_i}$ | 73.24$_{(\uparrow 0.00)}$ | 70.87$_{(\downarrow 0.44)}$ | 65.16$_{(\downarrow 0.65)}$ | 65.16$_{(\uparrow 0.42)}$ |
| HSTEC | 72.85 | 69.49 | 65.75 | 65.50 |
| + $\mathbf{L}_{t_i}$ | 73.83$_{(\uparrow 0.98)}$ | 71.45$_{(\uparrow 1.96)}$ | 66.83$_{(\uparrow 1.08)}$ | 65.80$_{(\uparrow 0.30)}$ |
| NaiveCat | 69.92 | 66.72 | 65.33 | 64.74 |
| + $\mathbf{L}_{t_i}$ | 73.44$_{(\uparrow 3.52)}$ | 71.49$_{(\uparrow 4.77)}$ | 65.46$_{(\uparrow 0.13)}$ | 64.12$_{(\downarrow 0.62)}$ |

Table 5: The experiment results (%) with different settings of learnable parameter $c$.

| Methods | $c$-parameter hypothesis | MVSA-Single | | MVSA-Multiple | |
|---|---|---|---|---|---|
| | | **Accuracy** | **F1** | **Accuracy** | **F1** |
| **CF-MSA** | Random | 73.24 | 72.97 | 64.50 | 61.61 |
| | Prior | 73.83 | 73.14 | 63.49 | 61.65 |
| | Uniform | 69.92 | 67.11 | 64.09 | 64.75 |
| | Non-uniform | 74.80 | 74.13 | 65.75 | 64.99 |

ample, compared to the OTE model (de Toledo & Marcacini, 2022), CF-MSA increases 2.93% and 1.61% in terms of accuracy and F1 scores on the MVSA-Single dataset when removing the text bias. These improvements demonstrate that the CF-MSA model can better explore reasonable modality information in text and image by introducing causal reasoning, especially the severe data bias of explicit text modality, thereby alleviating the impact of modal bias on multimodal sentiment prediction. Notable, when removing image bias and text-image Bias, the performance of the CF-MSA model remains at a relatively high level, although there is a slight decrease in its performance compared with removing text bias. The main reason is that the explicit text modality is easily influenced by words with sentiment polarity, while the image modality has less bias due to its rich semantic content. Overall, however, it can be observed that the proposed counterfactual model brings good results when different bias conditions are considered.

## 4.4 GENERALIZATION ABILITY

In order to further verify the generalization ability of our proposed CF-MSA, we conduct the extensive experiments in Table 3 on four existing state-of-the-art methods, i.e. OTE, CMAC, HSTEC, NaiveCat, with our proposed causal counterfactual inference strategies. It contains three situational conditions: removing text bias (i.e. w/o $T \rightarrow Y$), removing image bias (i.e. w/o $I \rightarrow Y$), and removing text-image biases (i.e. w/o $T \rightarrow Y$ and $I \rightarrow Y$). The performance of these models under different conditions, including accuracy and F1 scores, is summarized as shown in Table 3. From the experimental results in Table 3, we can observe that by introducing the causal inference mechanism, each model exhibits different degrees of performance improvement when dealing with the task of sentiment analysis under specific biases.

## 4.5 EFFECT OF NEW OPTIMISATION OBJECTIVE

After verifying the influence of modality biases on multimodal sentiment prediction, we further introduce a new optimisation objective $\mathcal{L}_{t_i}$ in the multimodal disambiguation methods, enabling the model to reduce the discrepancy between the $Z_{t,i^*,k^*}$ and $Z_{t^*,i,k^*}$ distributions during the optimisation process. To verify our proposed intermodal bias mutual elimination method, we conduct a series of ablation experiments for the new optimisation objective $\mathcal{L}_{t_i}$ on the four existing multimodal sentiment prediction methods in Table 4. The significant performance improvement proves the effectiveness of our proposed objective function. It reveals that we effectively mitigate the data bias in sentiment prediction performance due to single modality bias.

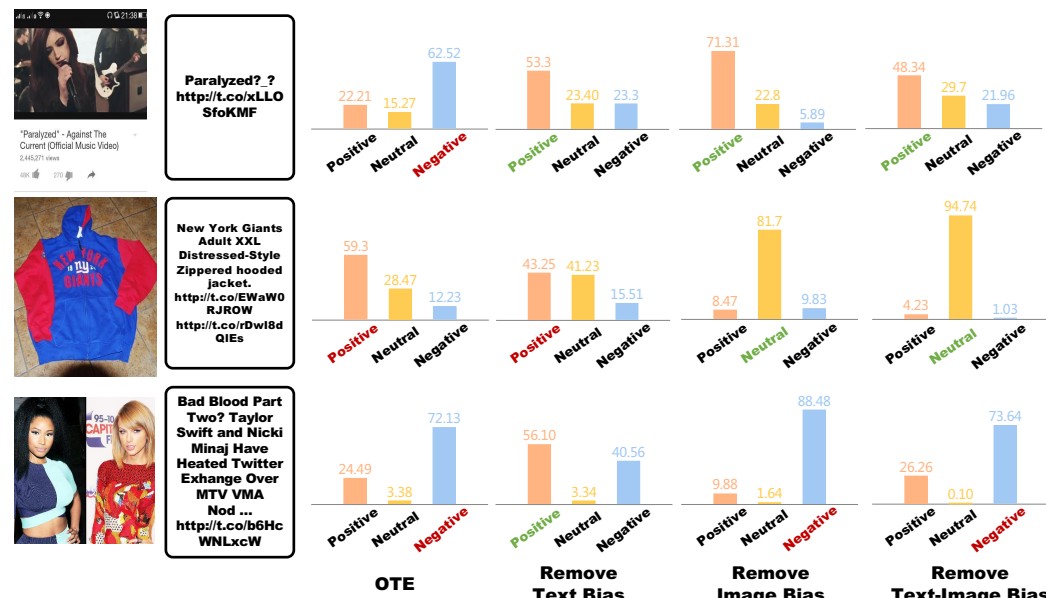

Figure 4: Qualitative analysis of test set examples. The first probability distribution chart is the prediction result of the traditional model, the second is the prediction result after removing the text bias, the third is the prediction result after removing the image bias, and the fourth is the result after removing both image and text biases. Red indicates a label that was incorrectly predicted, while green indicates a label that was correctly predicted.

### 4.6 EFFECT OF PARAMETER SETTINGS

To verify the role of learnable parameter $c$ in multimodal sentiment analysis, we conduct ablation experiments to assess the impact of different $c$-value settings on model performance, here including four assumptions, namely, random, prior knowledge from dataset statistic, uniformly distributed (with only one learnable parameter $c$), and non-uniformly distributed Niu et al. (2021) (differentiated treatment with multiple learnable parameters). The results are shown in Table 5. Analysis of the data in the above table yields that in most cases, the a priori and non-uniform distribution assumptions lead to a better boost in model performance. Thus, the non-uniform distribution assumption is necessary here based on the experimental model.

### 4.7 QUALITATIVE RESULTS

As shown in Figure 4, without removing modality bias, the model is prone to be influenced by explicit affective words in the text during prediction, leading to bias. For example, in the first example, the word "paralysis" in the text generally has a negative affective tendency, so the model tends to predict the affective of this text-image pair as "negative". However, in reality, the image expresses positive emotion, so the model gives an incorrect prediction. After bias removal, the model can more accurately capture the emotion expressed in the image and correctly classify the instance as "positive". This further shows that by removing modality bias, the model can more accurately fuse multimodal information and avoid misclassifications caused by the emotional bias of a single modality.

## 5 CONCLUSION

This paper, based on traditional multimodal sentiment classification, a causal reasoning method is innovatively introduced, and a counterfactual multimodal sentiment analysis framework (CF-MSA) is proposed. This framework effectively addresses the problem of modality bias in multimodal sentiment analysis through counterfactual reasoning. Experimental results show that CF-MSA is significantly superior to traditional analysis methods in the task of multimodal sentiment classification, demonstrating its superior generalization ability and performance advantages.

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

# APPENDIX

## A CONCEPTS AND THEORIES RELATED TO CAUSAL REASONING

In this section, we introduce the basic theory and related methods of causal reasoning, with special emphasis on the application of counterfactual ideas.

**Causal graph** is a directed acyclic graph (DAG) representing causal relationships between variables, notated as G={V, E}, where V denotes the set of variables and E denotes the set of causal relationships. For example, if we are going to study the relationship between a student's test score S and his level of effort in studying at school, E, and the amount of homework, H, we can construct a causal diagram as shown in the Figure 5, where the causal diagram contains three variables, E has a direct effect relationship on S, while H is an intermediate.

**Counterfactual representation** to facilitate the understanding of what may be a potential outcome in reality in the counterfactual strategy, there are symbols set up here for counterfactual representation. Where in the exposition of this paper uppercase letters are used to symbolically represent the variables themselves, while the corresponding lowercase letters mark the specific values of these variables in the actual observations. For example, $S_{e,h} = S(E = e, H = h)$ denotes the value of $S$ when $E$ is set to $e$ and $H$ is set to $h$. In practice, we usually have a counterfactual representation of a variable as a function of its value. In practice, we usually have $h = H_e = H(E = e)$, implying that $H$ is the value at which $E$ is set to $e$. As shown in figure 5, there exist the following four counterfactual case representations, and it is important to note here that only in the counterfactual world can two cases of taking values occur.

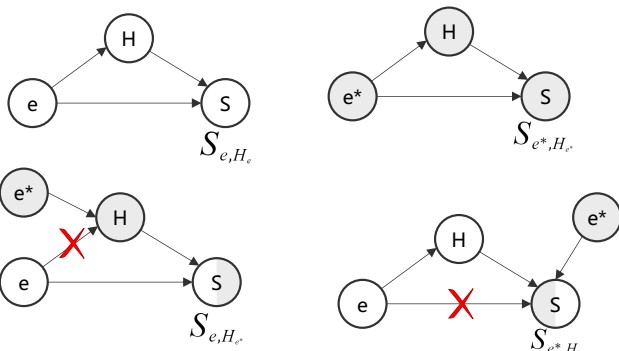

Figure 5: Cause-and-effect diagram examples and counterfactual situations.

**Causal effects** measure the difference in potential outcomes of a particular variable (e.g., test scores) under two different conditions (e.g., different levels of academic effort or amount of homework). For the example of a student's test score, $S$, consider two conditions: one in which the student exerts a certain level of effort, $E = e$, and completes a specific amount of homework, $H = h$, and the other at a different level of effort, $E = e^*$, and the amount of homework, $H = h^*$. The total effect (TE) of study effort and amount of homework on a student's test score can be defined as the difference between the potential outcomes of test scores in these two cases:

$$TE = S_{e,H_e} - S_{e^*,H_{e^*}} \tag{25}$$

Here $S_{e,H_e}$ is the potential value of the test score when the study effort is $e$, and $S_{e^*,H_{e^*}}$ is the potential value when the study effort is $e^*$.

The natural direct effect (NDE) measures the direct effect of the independent variable (study effort $E$) on the dependent variable (test score $S$) after controlling for the mediating variable (amount of homework $H$). In other words, the NDE reflects the change in $S$ brought about by a change in $E$ under the condition that the mediator variable remains at the no-intervention value $H_{e^*}$:

$$NDE = S_{e,H_{e^*}} - S_{e^*,H_{e^*}} \tag{26}$$

The total indirect effect (TIE), on the other hand, captures the effect that a change in the mediating variable (amount of homework H) has on S if E changes. It is the difference between the total effect

(TE) and the natural direct effect (NDE):

$$TIE = TE - NDE = S_{e,H_e} - S_{e,H_{e^*}} \tag{27}$$

# B  IMPLEMENT SETTINGS

In this section, we detail the implementation settings, including the model architecture, hyperparameter configurations, and training procedures used in our experiments.

## B.1  MODEL ARCHITECTURE

The model is a multi-modal framework that fuses information from both text and image inputs. It consists of three main components:

- **Text Encoder**: We utilize a pre-trained `RoBERTa-base` model for encoding the text input. The text encoder takes tokenized text sequences and their corresponding attention masks as input. The parameters of the RoBERTa model are partially fine-tuned with a learning rate of $5 \times 10^{-6}$.

- **Image Encoder**: For image encoding, we employ a ResNet-101 architecture pre-trained on ImageNet. The input images are resized to $224 \times 224$ and processed through the network, whose parameters are partially fine-tuned with a learning rate of $5 \times 10^{-6}$.

- **Fusion Module**: After extracting the text and image features, they are fused using a multi-head attention mechanism. This fusion module consists of 8 attention heads with a dropout rate of 0.4. The fused features are then passed through fully connected layers to generate the final predictions.

## B.2  HYPERPARAMETERS

The hyperparameters used in the model are as follows:

- **Text Encoder Learning Rate**: $5 \times 10^{-6}$
- **Image Encoder Learning Rate**: $5 \times 10^{-6}$
- **Fusion Module Learning Rate**: $3 \times 10^{-3}$
- **Weight Decay**: 0 for all components
- **Batch Size**: 16 for training and validation, 8 for testing
- **Number of Epochs**: 20
- **Dropout Rate**: 0.2 for the text and image encoders, 0.4 for the fusion module
- **Number of Attention Heads**: 8
- **Loss Weights**: The loss function is weighted with the class distribution [1.68, 9.3, 3.36] to account for the imbalanced dataset.

## B.3  TRAINING PROCEDURE

The model is trained for 20 epochs using the AdamW optimizer. We use two separate optimizers:

- **AdamW for main parameters**: This optimizer is used for fine-tuning the text encoder (RoBERTa), image encoder (ResNet), and fusion module parameters. The learning rate is set to $3 \times 10^{-3}$.

- **AdamW for specific parameters**: A separate optimizer is used to update the KL divergence-related parameters (`c, c2, c3, c4`) with a learning rate of $1 \times 10^{-5}$.

During each training iteration, the KL divergence loss is computed first and backpropagated using the specific optimizer, followed by the main cross-entropy loss for classification tasks. Gradients are computed and updated after each batch, and logging is performed to track the progress of the KL parameters over the training process.

### B.4 EVALUATION

For evaluation, we use accuracy metrics based on classification results for each batch during the validation phase. Additionally, a detailed classification report, including precision, recall, and F1-score, is generated using scikit-learn's `classification_report` function. The validation and test sets are processed with the same batch size settings as in training, and the model's predictions are compared against the ground-truth labels for performance analysis.

### B.5 HARDWARE AND FRAMEWORK

The implementation is done using PyTorch, with GPU acceleration enabled. The model is trained on an NVIDIA GPU using CUDA. The use of tqdm provides real-time training progress updates, and `torch.autograd.set_detect_anomaly(True)` is employed to detect potential issues during backpropagation.

