# OpenReview forum: "Multimodal Sentiment Analysis Based on Causal Reasoning"
_ICLR.cc/2025/Conference — Submitted to ICLR 2025_

### Official Review · Reviewer_rsbT · 2024-10-18

**Soundness:** 2
**Presentation:** 3
**Contribution:** 1
**Rating:** 3
**Confidence:** 5

**Summary:**

The paper presents a novel framework called CounterFactual Multimodal Sentiment Analysis (CF-MSA), addressing the issue of unimodal data bias in multimodal sentiment analysis, particularly focusing on the disparity between textual and image data. The authors leverage causal counterfactual inference to disentangle and mitigate the direct effects of biased unimodal data on final sentiment classification. The framework introduces differentiated treatment variables across modalities to ensure that multimodal sentiment analysis benefits from both complementarity and reduced bias between text and image data.

**Strengths:**

1. The framework successfully mitigates unimodal biases and ensures effective integration of heterogeneous modalities.
2. The paper is well-organized and easy to follow.
3. Strong experimental results, showing significant improvements over existing methods on multiple datasets.

**Weaknesses:**

1. The paper lacks novelty as causal inference methods have already been applied in other multimodal tasks ([1] Counterfactual VQA: A Cause-Effect Look at Language Bias and [2] Counterfactual Reasoning for Out-of-distribution Multimodal Sentiment Analysis), and the approach presented does not introduce significant innovations specifically tailored to the multimodal sentiment analysis (MSA) task.
2. The first claimed contribution is not valid, as there are already existing works that introduce causal effects into multimodal sentiment analysis frameworks [Counterfactual Reasoning for Out-of-distribution Multimodal Sentiment Analysis]. Furthermore, the authors fail to acknowledge and cite these relevant prior works, which undermines the originality of their contribution.
3. The model used in the experiments is overly simplistic, relying on a standard BERT-based text encoder, a ResNet-based image encoder, and a multi-head attention fusion module.

**Questions:**

See weakness

---

### Official Review · Reviewer_GQW8 · 2024-10-26

**Soundness:** 2
**Presentation:** 2
**Contribution:** 1
**Rating:** 3
**Confidence:** 5

**Summary:**

This paper proposes a multimodal sentiment analysis framework based on causal reasoning, CF-MSA (CounterFactual Multimodal Sentiment Analysis). This method reduces single-modal bias through causal counterfactual reasoning, thereby improving the accuracy of sentiment classification. Different from traditional methods, CF-MSA analyzes the direct impact of a single modality on sentiment classification by simulating the scenario in the absence of specific modal information, and eliminates modal bias in sentiment judgment. The method achieves the latest performance on the MVSA-Single and MVSA-Multiple datasets, which proves the effectiveness of the framework in the debiasing process.

**Strengths:**

1. The authors propose a modal-unbiased approach to solve the problem of MSA.

2. The proposed method is simple and easy to understand.

**Weaknesses:**

1. Lack of motivation and evidence: In line 57, the author's motivation should be supplemented with evidence, such as examples or citations.

2. I noticed that in the dataset, images and text have corresponding labels, why don't you use the corresponding labels in equation 20 for training.

3. The authors believe that the proposed method is unbiased, but it is not convincing to use only the MVSA dataset published in 2017, and more datasets should be added to validate, such as TumEmo [1], HFM [2], TWITTER-15 [6], TWITTER-17 [6], or constructing datasets.

4. Lack of improvement: Baseline comparisons are missing, and the baseline methods compared by the authors are 3 years ago and should be supplemented with more recent methods. For example: CLMLF [3], MVCN [4], MDSE [5].

5. I've noticed why this happens when the text bias is removed for optimal results, and the text and image biases are removed for suboptimal results？

6. The models and parameters presented by the author in the text do not correspond to the details in Appendix B.1 and should be modified.

7. The code should be open source to verify the performance of the model.

[1]. Image-Text Multimodal Emotion Classification via Multi-View Attentional Network. IEEE Transactions on Multimedia.

[2]. Multi-Modal Sarcasm Detection in Twitter with Hierarchical Fusion Model. ACL, 2019.

[3]. CLMLF:A Contrastive Learning and Multi-Layer Fusion Method for Multimodal Sentiment Detection. NAACL, 2022.

[4]. Tackling Modality Heterogeneity with Multi-View Calibration Network for Multimodal Sentiment Detection. ACL, 2023.

[5]. Modality-Dependent Sentiments Exploring for Multi-Modal Sentiment Classification. ICASSP, 2024.

[6]. Adapting bert for targetoriented multimodal sentiment classification. IJCAI, 2019.

**Questions:**

None

**Details Of Ethics Concerns:**

N

---

### Official Review · Reviewer_VGgH · 2024-10-29

**Soundness:** 2
**Presentation:** 2
**Contribution:** 1
**Rating:** 3
**Confidence:** 5

**Summary:**

This paper proposes a novel approach to multimodal sentiment analysis through a causal reasoning framework called CF-MSA. By leveraging counterfactual causal inference, CF-MSA aims to simulate scenarios where each modality’s influence is independently assessed, allowing the model to identify the true causal factors behind sentiment classification and make unbiased predictions. The paper demonstrates the effectiveness of CF-MSA on two datasets, MVSA-Single and MVSA-Multiple, where it outperforms existing models, achieving new state-of-the-art results.

**Strengths:**

- The paper communicates its complex causal methodology effectively, using figures to illustrate key concepts such as the counterfactual reasoning workflow and bias elimination process (e.g., Figure 3). Terms are clearly defined, and formulas are provided with sufficient context, which aids in understanding the causal mechanics behind the approach.

- The proposed CF-MSA framework has significant implications for improving multimodal sentiment analysis by offering a structured method to counteract modality biases.

**Weaknesses:**

- The paper’s methodology closely resembles existing multimodal works using causal reasoning, such as [1 - 3], but lacks a clear discussion distinguishing its contributions.

- The paper's contributions appear overstated, as the authors claim, “We are the first to introduce causal effects into the multimodal sentiment analysis framework.” However, prior work [1] has already applied causal reasoning within multimodal sentiment analysis.

- The handling of the Intermodal Discrepancy Treatment module is identical to that in [1], with Lines 286-291 mirroring descriptions found in the cited work. Additionally, the Integration Strategy module employs the SUM fusion approach from [1] without citation.

- The Intermodal Bias Mutual Elimination Optimization objective aims to balance the distribution by eliminating extremes, yet lacks direct evidence through distribution visualization.

- The study lacks experiments on closed-source LVLMs (e.g., GPT-4, Claude 3.5).

- Despite claiming model-agnostic applicability, the paper does not evaluate this method on open-source LVLMs like LLaVA-1.5, MiniGPT-4, and mPLUG-Owl2.

[1] Niu Y, Tang K, Zhang H, et al. Counterfactual VQA: A cause-effect look at language bias[C]//Proceedings of the IEEE/CVF conference on computer vision and pattern recognition. 2021: 12700-12710.

[2] Chen Z, Hu L, Li W, et al. Causal intervention and counterfactual reasoning for multi-modal fake news detection[C]//Proceedings of the 61st Annual Meeting of the Association for Computational Linguistics (Volume 1: Long Papers). 2023: 627-638.

[3] Sun T, Wang W, Jing L, et al. Counterfactual reasoning for out-of-distribution multimodal sentiment analysis[C]//Proceedings of the 30th ACM International Conference on Multimedia. 2022: 15-23.

**Questions:**

N/A

---

### Official Review · Reviewer_rLmn · 2024-11-01

**Soundness:** 2
**Presentation:** 2
**Contribution:** 2
**Rating:** 3
**Confidence:** 4

**Summary:**

This paper proposes new framework based on causal counterfactual inference for the multimodal sentiment analysis task, effectively addressing the issue of modality bias in multimodal sentiment analysis and demonstrating high generalization capabilities.

**Strengths:**

1. This paper presents a newl framework based on causal counterfactual reasoning, offering some insights for multimodal sentiment analysis task.
2. Many experiments have been conducted, and the implementation detail is relatively comprehensive.

**Weaknesses:**

1. In Section 4.1, this paper mentions that this paper removes image-text pairs with inconsistent modality labels, but in Section 4.7, this paper provides an example where the text emotion label is negative while the image emotion label is positive, which seems contradictory. Additionally, if the labels for both text and image are consistent, is it still worth eliminating modality bias?
2. I notice that the methods selected for the experiments are all from (or before) 2022. Have there been no new methods in the field of multimodal sentiment analysis in the past two years?
3. The novelty of both the proposed approach and the addressed task is relatively limited.
4. The illustration of the causal inference in the proposed approach should be more clear.

**Questions:**

Please see Weaknesses.

---

### Meta-Review · Area_Chair_RFT9 · 2024-12-20

**Metareview:**

The authors did not provide responses to the reviews, and several major concerns including unclear motivation, lack of technical contributions, and absence of baseline remain unaddressed. As a result, we recommend rejecting this paper in the meta-review.

**Additional Comments On Reviewer Discussion:**

Several major concerns including unclear motivation, lack of technical contributions, and absence of baselines remain unaddressed.

---

### Decision · Program_Chairs · 2025-01-22

Reject